# Research on Dynamic Modeling Method and Flying Gait Characteristics of Quadruped Robots with Flexible Spines

**DOI:** 10.3390/biomimetics9030132

**Published:** 2024-02-21

**Authors:** Lei Jiang, Zhongqi Xu, Tinglong Zheng, Xiuli Zhang, Jianhua Yang

**Affiliations:** 1School of Computer Science and Technology, Zhejiang University, Hangzhou 310027, China; 11921167@zju.edu.cn; 2School of Mechanical, Electronic and Control Engineering, Beijing Jiaotong University, Beijing 100044, China; xzq_brave@163.com (Z.X.); 23111370@bjtu.edu.cn (T.Z.)

**Keywords:** quadruped robots, flexible spine, dynamic model, flight phase motion

## Abstract

In recent years, both domestic and international research on quadruped robots has advanced towards high dynamics and agility, with a focus on high-speed locomotion as a representative motion in high-dynamic activities. Quadruped animals like cheetahs exhibit high-speed running capabilities, attributed to the indispensable role played by their flexible spines during the flight phase motion. This paper establishes dynamic models of flexible spinal quadruped robots with different degrees of simplification, providing a parameterized description of the flight phase motion for both rigid-trunk and flexible-spine quadruped robots. By setting different initial values for the spine joint and calculating the flight phase results for both types of robots at various initial velocities, the study compares and analyzes the impact of a flexible spine on the flight phase motion of quadruped robots. Through comparative experiments, the research aims to validate the influence of a flexible spine during the flight phase motion, providing insights into how spine flexibility affects the flight phase motion of quadruped robots.

## 1. Introduction

The high adaptability and agile movement characteristics of quadruped robots in unstructured environments make them significant in specific applications and research areas, such as earthquake or fire rescue operations, wilderness military support, indoor household services, and educational entertainment. Consequently, quadruped robots have become a focal point in the field of robotics. Currently, research on quadruped robots mostly revolves around rigid-bodied quadruped robots [1,2,3,4,5], with a primary focus on bio-inspired leg structures and the study of gait generation and coordinated control among the four limbs. In practical quadruped mammals, however, the flexible spine joints play a crucial role in achieving high-speed, agile, and stable motion.

In recent years, there have been two main types of spine structures adopted for quadruped robots, namely articulated spines [6,7,8] and integral compliant spines [9,10,11,12,13,14]. Although the mechanisms by which a flexible spine influences dynamic performance remain elusive, domestic and international research suggests that a flexible spine can enhance a robot’s movement speed, increase stride length, enable elastic energy storage, and reduce energy consumption [6,7,8,10,11,12,13,14]. Stoch2, a quadruped robot designed by Shounak Bhattacharya and colleagues at the Indian Institute of Science, features a dual-joint spine. Drawing inspiration from the running motion of cheetahs, this robot utilizes deep reinforcement learning to generate joint control signals. The results indicate a significant improvement in the robot’s running speed and stride length and a reduction in locomotion costs facilitated by the flexible spine [6]. Sugoi-Neco, a quadruped robot with a passive single-joint spine designed by Ryosuke Kawasaki and his team at the University of Electro-Communications in Japan, employs a spine joint with torsional springs for storing elastic potential energy. The findings suggest that the energy stored in the flexible spine allows the robot to generate larger horizontal and vertical impulses during the support phase, enabling it to achieve greater leaps [8]. The MIT Cheetah I robot, designed by Sangbae Kim’s team at the Massachusetts Institute of Technology, features a spine composed of rigid vertebrae interspersed with polyurethane rubber rings. The spine is coupled to the hind legs by connecting ropes, utilizing a differential mechanism linked to the hip joint motors. This coupling achieves synchronized motion between the spine and hind legs. The robot ultimately achieved a running speed of 22 km/h in a trot gait on a treadmill, with the flexible spine contributing to reducing the robot’s locomotion costs [9,10]. The Kitty robot, designed by Zhao Qian and colleagues at the University of Zurich, has a spine consisting of ABS-made cross-shaped vertebrae alternately connected with silicone blocks. Four servos drive ropes in four directions to bend the spine. It was found that the robot exhibits the fastest forward speed when the virtual spine joint is located on the posterior side of the spine [11]. Canid and Inu are two quadruped robots with integral compliant spines developed with the involvement of Daniel E. Koditschek’s team at the University of Pennsylvania. The spines of these robots directly use carbon fiber plate springs and glass fiber leaf spring plates, respectively, as the main body of the spine. Motors rotate to actively control the spine by pulling ropes or belts passing through it. Experimental results indicate that a flexible spine can increase the average speed of quadruped robots by 0.4 m/s, decrease the body pitch angle by 17%, and reduce the average vertical height of the center of mass by approximately 3 cm [12,13]. The quadruped robot with a passive integral compliant spine, Fanari, designed by Hasan H. Kani and colleagues at the University of Tehran in Iran features a spine composed of multiple organic glass segments parallelly connected with linear springs. In experiments where the robot sprints along a slope without any external energy, it was observed that the spine, capable of bending in both upward and downward directions, enables the robot to achieve faster and more stable movement compared to spines that can only bend upward or rigid spines [14].

Due to the additional degrees of freedom introduced by a flexible spine in quadruped robots and the necessity for the motion of the spine to be coordinated with the movement of the remaining degrees of freedom, without mutual inhibition, numerous researchers worldwide are currently exploring the mechanisms underlying the impact of a flexible spine on various performance metrics of quadruped robots. They are establishing corresponding dynamic models and conducting motion characteristic analyses [15,16,17,18,19,20,21,22]. However, there is limited research investigating the influence of a flexible spine on the flight phase motion of quadruped robots. In 2015, Soroush Maleki and colleagues from the University of Tehran addressed quadruped robots with flexible spines. Using the Lagrange method, they formulated motion equations and, based on the model, employed a genetic algorithm for gait design optimization, ultimately reducing the system’s energy consumption [16]. In 2018, Yesilevskiy proposed a quadruped robot model with distributed mass. By comparing the dynamic characteristics of rigid-spine and flexible-spine quadruped robots, it was found that, for asymmetric gaits, introducing a flexible spine can increase the maximum running speed, extend the stride, and improve energy utilization efficiency through the energy storage and conversion functions of the spine [21]. In 2020, Zeng Shun and colleagues from Wuhan University of Technology investigated the influence of the center of mass position of spine segments on the dynamic performance of the bound gait in quadruped robots. They established a passive simplified model in the sagittal plane and treated the center of mass position of the spine segment as an independent variable. The results indicated that when the center of mass of the spine segment is closer to the hip joint, the bound gait cycle and stride will be longer. Conversely, when the center of mass is closer to the midpoint of the spine segment, the fluctuation in the robot’s center of gravity and the ground reaction force will be smaller [22].

In this paper, we concentrate on quadruped robots with a flexible spine and formulate dynamic models with varying degrees of simplification. Drawing inspiration from the observed flight phase motion of cheetahs during high-speed galloping, we introduce parameterized descriptions of the flight phase process for both rigid-bodied and flexible-spine quadruped robots. To assess the impact of a flexible spine on the flight phase, we assign different initial values to the spine joint and compute the flight phase outcomes for both types of quadruped robots at various initial speeds. Through comparative experiments, we affirm the influence of a flexible spine during the flight phase. The primary contributions of this paper include the establishment of dynamic models of different degrees and the parameterized description of the flight phase for quadruped robots. The study validates the influence of a flexible spine during the flight phase, inspired by observations of cheetahs in high-speed galloping.

The remainder of the paper is organized as follows: In Section 2, the dynamic models of quadruped robots with different degrees of simplification are introduced. Section 3 provides a parameterized description of the flight phase process for both rigid-bodied and flexible-spine quadruped robots. Section 4 compares the impact of rigid-bodied and flexible-spine configurations on the flight phase of quadruped robots and analyzes the results of this comparison. Finally, Section 5 concludes the paper and discusses future work.

## 2. Quadruped Robot Model

### 2.1. Dynamic Models of Varying Degrees of Simplification

We have established a quadruped robot model with a spine, as illustrated in Figure 1. This model consists of a body with a spine and four legs, each leg comprising a thigh and a shank. The model has 9 degrees of freedom, with each leg having two degrees of freedom: knee joint pitch and hip joint pitch. There is no hip roll degree of freedom, and there is one spine joint degree of freedom. The parameters of this model are presented in Table 1.

#### 2.1.1. Point Mass Model

Due to the prevalent design of low inertia in the leg structure and the centralized arrangement of drive motors at the hip joints in contemporary quadruped robots [2], this simplified model considers only the combined center of mass of the two trunk segments. The model depicted in Figure 1 is thus simplified to a point mass model, where the mass possesses no volume or shape considerations, as shown in Figure 2a. The external forces acting on it include constraint forces from the ground and the gravitational forces of the two trunk segments. The forces can be entirely translated to a single point, forming a concurrent force system. The position vector of the modeled point mass relative to the world coordinate system can be expressed as
(1)$$r_{c}=\frac{m_{bB}r_{bB}+m_{bF}r_{bF}}{m_{bB}+m_{bF}}=\frac{m_{bB}r_{bB}+m_{bF}r_{bF}}{m_{0}}$$
where $m_{0}$ represents the mass of the robot’s trunk and $r_{c}$ is the position vector of the point mass in the world coordinate system. Since forces only influence changes in the position of the point mass, based on Newton’s Second Law, the dynamic equation for this point mass can be derived as
(2)$$m_{0}g+\sum_{i=1}^{4}\lambda_{i}=m_{0}\cdot {\ddot{r}}_{c}$$
where $\lambda_{i}$ represents the constraint force applied by the ground at the end of each leg of the robot.

#### 2.1.2. Single-Rigid-Body Model

The simplification in the single-rigid-body model involves neglecting the dynamic characteristics of the robot’s legs, considering the robot’s torso as a single rigid body, and assuming the robot has only a torso and foot ends. The model is illustrated in Figure 2b, where the coordinate system $\left\{B\right\}$ represents the body-attached coordinate system.

The external forces acting on the four-legged robot with a single rigid body consist of the body gravity and the ground reaction forces at the foot ends. For the translational motion of the rigid body, the translational motion equation of this model can be derived from Euler’s first law of motion as follows:(3)m0g+∑i=14λiW=m0⋅r¨cW

For the rotational motion of the rigid body, choosing the body’s center of mass as the reference point, the external torque on the rigid body can be obtained from Euler’s second law as
(4)MW=∑i=14rctiW×λiW+03×1×m0⋅gW
where M represents the external torque acting on the rigid body.

The angular momentum L of the rigid body, rotating about an axis passing through its center of mass and perpendicular to the plane of the paper, is equal to the product of the rotational inertia J and the angular velocity $\omega_{WB}$ of the rigid body relative to the world coordinate system. This relationship is expressed in Equation (5). Here, the rotational inertia J about the axis passing through the center of mass and perpendicular to the paper can be calculated as $J=\frac{1}{12}\cdot m_{0}\cdot {\left(l_{bB}+l_{bF}\right)}^{2}$, where $l_{bB}$ is the distance from the center of mass to the back foot and $l_{bF}$ is the distance from the center of mass to the front foot.
(5)LW=J•ωWBW

Thus, Equations (3) and (6) together form the dynamic equations of the single-rigid-body model.

According to Euler’s second law, combining Equations (4) and (5), we can express the equations of motion for the rigid body undergoing rotational motion about a fixed axis in the plane as
(6)∑i=14rctiW×λiW+03×1×m0⋅gW=dWLdt=J⋅dwWBWdt

#### 2.1.3. Planar Multiple-Rigid-Body Model

Unlike the point mass model and the single-rigid-body model, the planar multiple-rigid-body model considers the legs explicitly and precisely accounts for the motion of each body in the robot. As shown in Figure 3, a planar multiple-rigid-body model is established, treating the quadruped robot as a floating base object. The motion of the floating base with respect to the world coordinate system is achieved through interactions between the foot ends and the ground, without direct actuation. This model is formulated using the first kind of Lagrange equations, and according to the first kind of Lagrange formalism, dynamic equations are derived for the abstracted 12-DOF model depicted in Figure 3.
(7)$$\frac{d}{dt}(\frac{\partial L\;}{\partial \;{\dot{q}}_{k}})-\frac{\partial L\;}{\partial \;q_{k}}=Q_{k}+\Gamma_{k}\;\;\;\;\;\;\;\left(k=1,\;2,\cdots ,12\right)$$

To express each term of Equation (7), we write the equation for each generalized coordinate’s motion. Then, we organize these 12 equations into matrix form, which can be found in Appendix A. Finally, we provide a concise representation of the matrix form.
(8)$$M\left(q\right)\ddot{q}+C(q,\dot{q})+G(q)=B\cdot u+J^{T}\lambda$$
where the coefficient matrix $M\left(q\right)$ is referred to as the system’s inertia matrix, $C\left(q,\dot{q}\right)$ is the Coriolis and centrifugal vector matrix of the robot, $G\left(q\right)$ is the gravity vector matrix of the robot, and B is the selection matrix. The 8-dimensional input torque vector u, representing the torques applied at each joint, is transformed into a 12-dimensional column vector. The first four elements of the vector are zeros, and the last eight elements are equal to the input torque vector u. J is the stacked Jacobian matrix of the ground-contacted foot ends.

## 3. Quadruped Robot Flight Phase Motion Parameterization

### 3.1. Flight Phase Motion Simplification

To simplify the analysis, this paper chooses the process starting from the cheetah’s hind legs touching the ground, through the extension of the trunk, to the realization of the large flight phase, and finally to the front legs touching the ground as a reference [23]. According to the force and motion characteristics of the cheetah, the whole process is divided into take-off motion and flight phase motion, which is subjected to gravity and ground reaction force during the take-off motion, and only gravity during the flight phase motion, as shown in Figure 4.

Referring to the motion characteristics of the cheetah’s large flight phase process in Figure 4, the flight phase process of the quadruped robot with a rigid torso and flexible spine is simplified in Figure 5 and Figure 6. In the simplified process, the left and right pairs of legs of the robot are set to have the same motion phase, and in the take-off motion, only the robot’s back leg exerts force, the angle of the joints of the front leg does not change, and the angle of each joint of the front and back legs is set to remain constant until the end of the flight phase motion. The angles of the front and back legs of the robot remain unchanged until the end of the flight phase motion, and the angles of the spine joints are set to change only during the take-off motion, while the angles remain fixed during the flight phase motion.

### 3.2. Rigid-Torso Quadruped Robot Flight Phase Motion Parameterization

The flight phase motion of a rigid-torso quadruped robot can be decomposed into translational motion of the center of mass and fixed-axis rotation over the center of mass and perpendicular to the plane, as shown in Figure 7.

In the take-off motion process, only by the vertical direction of gravity, the whole process is equivalent to the oblique throw movement, with the take-off point as a benchmark, and is combined with the expected process to achieve the flight phase trajectory height of h. The landing point (LA) coordinates for $\left(x_{LA},\;y_{LA}\right)$ can be known as the shape of the parabola, and the center of mass in the horizontal and vertical direction of the speed and position change trajectory, the time of the flight phase, the angle of the landing point can be obtained as follows:(9)$$\left\{\begin{array}{l} v_{cx\_FL}\left(t\right)=v_{LOx},\;\;\;\;\;\;\;\;v_{cy\_FL}\left(t\right)=v_{LOy}-gt\\ x_{c\_FL}\left(t\right)=v_{LOx}•t,\;\;\;\;y_{c\_FL}\left(t\right)=v_{LOy}t-\frac{1}{2}gt^{2}\\ \Delta t_{FL}=\frac{\sqrt{2gh}+\sqrt{2gh-2gy_{LA}}}{g}\\ \theta_{LA}=\theta_{LO}+\omega_{LO}•\Delta t_{FL} \end{array}\right.$$

In the flight phase motion process, according to Ludovic D. Maes et al. on the biology of dogs, it was shown that there is a certain pattern between the gait frequency f as well as the loading factor β and the horizontal velocity $v_{x}$, which corresponds to the following mathematical relationship [24]:(10)$$\left\{\begin{array}{l} f=1.314+0.762\ln v_{x}\\ \beta =0.652-0.21\ln v_{x} \end{array}\right.$$

The horizontal instantaneous velocity at the onset point is approximately equal to its average velocity during a gait cycle [23]. The duration $▵t_{TO}$ of the take-off motion can be obtained as
(11)$$▵t_{TO}=\frac{\beta }{f}=\frac{0.652-0.21\ln v_{STx}}{1.314+0.762\ln v_{STx}}$$

According to the momentum theorem, the horizontal and vertical ground reaction forces $\lambda_{x}\left(t\right)$ and $\lambda_{y}\left(t\right)$ applied to the robot during the take-off motion should satisfy the following conditions:(12)$$\int_{0}^{\frac{\beta }{f}}\lambda_{x}(t)\;dt=m(v_{LOx}-v_{STx})\;\Rightarrow \;\int_{0}^{\frac{\beta }{f}}\lambda_{x}(t)\;dt=\frac{mgx_{LA}}{\sqrt{2gh}+\sqrt{2gh-2gy_{LA}}}-mv_{STx}$$
(13)$$\int_{0}^{\frac{\beta }{f}}\lambda_{y}(t)-m_{0}g\;dt=m(v_{LOy}-v_{STy})\;\Rightarrow \;\int_{0}^{\frac{\beta }{f}}\lambda_{y}(t)\;dt=mg\frac{\beta }{f}+m(\sqrt{2gh}-v_{STy})$$

Based on the above conditions, it is still not possible to obtain specific trajectories of ground reaction forces. Referring to the trajectories of ground reaction force in gallop gait measured by Penny E. Hudson et al. for cheetahs and greyhounds running at high speed [25] and Rebecca M. Walter and David R. Carrier for adult dogs running at high speed [26], Fourier series were used to fit the trajectories of ground reaction forces over time.

For the vertical ground reaction force $\lambda_{y}\left(t\right)$, three points, $\left(0,0\right),\;\left(0.4\beta /f,\;\lambda_{y}^{*}\right),\;\left(\beta /f,\;0\right)$, are chosen as fitting points, and at the same time, the impulse constraints of Equation (13) should be satisfied, so the vertical ground reaction force trajectory $\lambda_{y}\left(t\right)$ meets the bionic characteristics and also meets the conditions of impulse in the take-off motion, as shown in Equation (14).
(14)$$\lambda_{y}(t)=a_{1}\sin(2\pi ft)+a_{2}\sin2(2\pi ft)$$
where $a_{1},a_{2}$ are both Fourier series coefficients.

Similarly, for the horizontal direction ground reaction force $\lambda_{x}\left(t\right)$, five points, $\left(0,0\right)$, $\left(0.14\beta /f,-\lambda_{x}^{*}\right),\;\left(0.28\beta /f,\;0\right),\;\left(0.8\beta /f,\;0.3\lambda_{x}^{*}\right),\;\left(\beta /f,\;0\right)$, are selected as the fitting points and at the same time satisfy Equation (12); then, the trajectory of the ground reaction force $\lambda_{x}\left(t\right)$ in the horizontal direction can be expressed as
(15)$$\begin{array}{c} \lambda_{x}(t)=b_{1}\sin(2\pi ft)+b_{2}\sin2(2\pi ft)+b_{3}\sin3(2\pi ft)\\ +b_{4}\sin4(2\pi ft)+b_{5}\cos5(2\pi ft) \end{array}$$
where $b_{1}\sim b_{5}$ are all Fourier series coefficients.

After determining the trajectory of the ground reaction force in the take-off motion, the trajectory of the center of mass in the process in terms of velocity and position over time can be determined, as well as the angular trajectory of the torso, as shown in Equation (16).
(16)$$\left\{\begin{array}{l} v_{cy\_TO}(t)=v_{STy}+\int_{0}^{t}a_{y\_TO}(t)\;dt=v_{STy}+\int_{0}^{t}\frac{\lambda_{y}(t)-m_{0}g}{m_{0}}\;dt\\ y_{c\_TO}(t)=y_{ST}+v_{STy}•t+\int_{0}^{t}\int_{0}^{t}\frac{\lambda_{y}(t)-m_{0}g}{m_{0}}\;dtdt\\ v_{cx\_TO}(t)=v_{STx}+\int_{0}^{t}a_{x\_TO}(t)\;dt=v_{STx}+\int_{0}^{t}\frac{\lambda_{x}(t)}{m_{0}}\;dt\\ x_{c\_TO}(t)=x_{ST}+v_{STx}•t+\int_{0}^{t}\int_{0}^{t}\frac{\lambda_{x}(t)}{m_{0}}\;dtdt\\ \theta_{TO}\left(t\right)=\theta_{ST}+\omega_{ST}•t+\int_{0}^{t}\int_{0}^{t}\frac{M\left(t\right)}{J}\;dtdt \end{array}\right.$$

In summary, the parameterization of the flying phase process of the rigid-torso quadruped robot is completed, as shown in Table 2.

### 3.3. Flexible-Spine Quadruped Robot Flight Phase Motion Parameterization

The overall center of mass of the torso composed of the centers of mass of the anterior and posterior torsos is selected as the object of the analyzed mass point, and the motion process of this mass point in the two-dimensional plane under a specific force is analyzed and described parametrically to represent the flight phase motion process of the flexible-spine quadruped robot in a similar manner, which is shown in the schematic diagram in Figure 8a. In order to describe the parameterized flight phase motion process of the flexible-spine quadruped robot, the trajectories of the spine joints in the whole motion process and the ground reaction force in the take-off motion process need to be clarified first.

#### 3.3.1. Spinal Joint Trajectory Planning

Referring to the changing rule of the flexible spine in cheetah running, obvious stretching action occurs in the spine joints only in the take-off motion, while the spine joint angle remains unchanged in the flight phase segment. The flexible spine joints are set to have the locked/unlocked function, and this locked/unlocked function of the flexible spine has been applied to the physical prototype in the research of other domestic counterparts, which is feasible [19]. The motion of the spinal joints is shown schematically in Figure 8b.

Let the angle of the spinal joints at the start point be $\varphi_{s0}$ and the stiffness of the joint torsion springs be a certain value. The anterior trunk will be subjected to a joint torque $M_{s}\left(t\right)$ due to the bending of the torsion springs, and $M_{s}\left(t\right)$ can be expressed as
(17)$$M_{s}\left(t\right)=-k_{s}•\varphi_{s}\left(t\right)=J_{bF}{\ddot{\varphi }}_{s}\left(t\right)=\frac{1}{3}m_{bF}{l^{2}}_{bF}•{\ddot{\varphi }}_{s}\left(t\right)$$

The solution corresponding to the spinal joint angle $\varphi_{s}\left(t\right)$ can be found as
(18)$$\varphi_{s}\left(t\right)=A\sin\left(\omega_{n}t+\varphi \right)$$
where A is the amplitude of the vibration and $\omega_{n}$ is the circular frequency, and it is easy to know that φ is the initial phase of the vibration. According to the initial condition, $\varphi_{s}\left(0\right)=\varphi_{s0}$, it can be solved that the amplitude of the vibration $A=\varphi_{s0}$ and the initial phase φ=π/2.

From the circular frequency $\omega_{n}$ of the system, the period T of the vibration can be written:(19)$$T=\frac{2\pi }{\omega_{n}}$$

The time of the angular motion of the joint is half of its vibration period and at the same time is equal to the time of the take-off motion $\Delta t_{TO\_flex}$. Let the initial speeds of the center of mass at its starting point in the horizontal and vertical directions be $\;v_{STx\_flex}$,$v_{STy\_flex}$. Based on Equation (10), the load factor $\beta_{flex}$ as well as the gait frequency $f_{flex}$ can be solved for the flexible-spine quadruped robot under this motion. Then,
(20)$$\frac{T}{2}=\sqrt{\frac{\pi^{2}m_{bF}{l_{bF}}^{2}}{3k_{s}}}=▵t_{TO\_flex}=\frac{\beta_{flex}}{f_{flex}}$$

The stiffness $k_{s}$ of the torsion spring of the spinal joint is
(21)$$k_{s}=\frac{\pi^{2}m_{bF}{l_{bF}}^{2}{f_{flex}}^{2}}{3{\beta_{flex}}^{2}}$$

The obtained stiffness $k_{s}$, amplitude A, and initial phase φ of the torsion spring of the spine joint are brought into Equation (18), and the derivation of the time is made, which means that the angular position of the spine joint and the trajectory of the angular velocity with time can be obtained:(22)$$\left\{\begin{array}{l} \varphi_{s}\left(t\right)=\left\{\begin{array}{c} {\varphi_{s}}_{0}\sin\left(\frac{f_{flex}}{\beta_{flex}}\pi t+\frac{\pi }{2}\right)\;\;\;\;\;\;\;\;\;\;\;\;t\in \left[0,\beta_{flex}/f_{flex}\right]\;\\ -{\varphi_{s}}_{0}\;\;\;\;\;\;\;\;\;\;\;\;t\in \left[\beta_{flex}/f_{flex},\beta_{flex}/f_{flex}+▵t_{FL\_flex}\right] \end{array}\right.\\ {\dot{\varphi }}_{s}\left(t\right)=\left\{\begin{array}{c} \frac{{\varphi_{s}}_{0}\pi f_{flex}}{\beta_{flex}}\cos\left(\frac{f_{flex}}{\beta_{flex}}\pi t+\frac{\pi }{2}\right)\;\;\;\;\;\;\;\;\;\;\;\;t\in \left[0,\beta_{flex}/f_{flex}\right]\\ 0\;\;\;\;\;\;\;\;\;\;\;\;\;\;\;\;\;\;\;\;\;\;\;t\in \left[\beta_{flex}/f_{flex},\beta_{flex}/f_{flex}+▵t_{FL\_flex}\right] \end{array}\right. \end{array}\right.$$

#### 3.3.2. Ground Reaction Force

In order to obtain the ground reaction force which is only affected by the flexible spine compared to the rigid-torso quadruped robot, the solution process is based on the ground reaction force of the rigid-torso quadruped robot. Firstly, according to the described flying phase process of the rigid-torso quadruped robot, the final obtained trajectories of the center of mass position, velocity, torso attitude angle, and angular velocity of the rigid torso over time are converted into the trajectories of the generalized position and generalized velocity variables over time in the complete dynamics model based on the geometric relationships. The geometric relationship between the above several physical quantities is schematically shown in Figure 9a, and the hip and knee angles of the front leg are set to maintain fixed values throughout the process. The specific values of both are set here as shown in Equation (23). After determining the trajectories of the variables of the generalized position, the generalized velocity is derived for the obtained generalized position with respect to time.
(23)$$\left\{\begin{array}{l} \alpha_{hFL\_rigid(hFR\_rigid)}=\frac{10}{9}\pi \\ \alpha_{kFL\_rigid(kFR\_rigid)}=\frac{7}{9}\pi \end{array}\right.$$

The input torque $u_{rigid}\left(t\right)$ for each joint of the rear leg of the transition segment can be found by bringing the ground reaction force of the rigid-torso quadruped robot based on the Fourier series fit with the derived generalized position and velocity trajectories into the planar multiple-rigid-body model (Equation (18)).

#### 3.3.3. Flight Phase Motion Parameterization

Since the position of the total center of mass is related to the position of the centers of mass of the front and rear trunks, the motions of the respective centers of mass of the front and rear trunks under the corresponding forces are first analyzed in the take-off motion. Subsequently, the motions of both are synthesized by mass distribution to obtain the motion of the total center of mass in the take-off motion. A schematic diagram of the force and the corresponding velocity analysis for the front and back trunks is shown in Figure 9b. The velocity $v_{bB}$ of the center of mass of the posterior torso is only affected by its own gravity $G_{bB}$ and the ground reaction force $\lambda_{flex}\left(t\right)$, while the velocity of the center of mass of the anterior torso is affected not only by its own gravity $G_{bF}$ and the ground reaction force $\lambda_{flex}\left(t\right)$, but also by the elastic force F corresponding to the release of flexible spine, and the two kinds of velocities are expressed as $v\prime_{\mathrm{bF}}$ and $v{*}_{bF}$, respectively. The final combined velocity of $v\prime_{bF}$ and $v{*}_{bF}$ is the velocity $v_{bF}$ of the anterior trunk center of mass in take-off motion.

Let the horizontal and vertical initial velocities of the robot’s total center of mass as well as the front and rear torsos at the starting point be $v_{STx\_flex}$, $v_{STy\_flex}$. Then, according to the ground reaction force $\lambda_{flex\_x}\left(t\right)$, $\lambda_{flex\_y}\left(t\right)$ of the flexible spine object obtained in the previous section, we can obtain the trajectory of the center of mass velocity of the front and rear torsos’ centers of mass in the presence of the ground reaction force with time:(24)$$\left\{\begin{array}{l} v_{bBx}\left(t\right)=v_{STx\_flex}+\int_{0}^{t}\frac{{\lambda_{flex}}_{\_x}(t)}{m_{0}}\;dt=v\prime_{bFx}\left(t\right)\\ v_{bBy}\left(t\right)=v_{STy\_flex}+\int_{0}^{t}\frac{\lambda_{flex\_y}(t)-m_{0}g}{m_{0}}\;dt=v\prime_{bFy}\left(t\right) \end{array}\right.$$

Since the quadrupedal robot with a flexible torso is reduced to a prime model, the angle between its rear side torso and the horizontal direction cannot be obtained, so this quantity is set equal to the torso attitude angle of the quadrupedal robot with a rigid torso. The following can be obtained:(25)$$\left\{\begin{array}{l} v{*}_{bFx}\left(t\right)=-\frac{1}{2}{\dot{\varphi }}_{s}(t)l_{bF}•\sin\left[\varphi_{s}\left(t\right)+\varphi_{b\_rigid}\left(t\right)\right]\\ v{*}_{bFy}\left(t\right)=\frac{1}{2}{\dot{\varphi }}_{s}(t)l_{bF}•\cos\left[\varphi_{s}\left(t\right)+\varphi_{b\_rigid}\left(t\right)\right] \end{array}\right.$$

The velocity and positional trajectories of the total center of mass of the torso in the horizontal and vertical directions can be obtained:(26)$$\left\{\begin{array}{l} v_{cx\_flex}\left(t\right)=\frac{m_{bB}•v_{bBx}\left(t\right)+m_{bF}•v_{bFx}\left(t\right)}{m_{0}}\;\;\;\;\;\;\;\;\;\;\;\;\;\;\;v_{cy\_flex}\left(t\right)=\frac{m_{bB}•v_{bBy}\left(t\right)+m_{bF}•v_{bFy}\left(t\right)}{m_{0}}\\ x_{c\_flex\_tran}\left(t\right)=x_{ST\_flex}+\int_{0}^{t}v_{cx\_flex}\left(t\right)\;dt\;\;\;\;\;\;y_{c\_flex\_tran}\left(t\right)=y_{ST\_flex}+\int_{0}^{t}v_{cy\_flex}\left(t\right)\;dt \end{array}\right.$$

Since the flexible spinal joints are in the “locked” state in the flight phase, the motion of the flexible torso center of mass in this process is the same as that of the rigid torso center of mass in the flight phase.

In summary, the parameterization of the flying phase process of the flexible-spine quadruped robot is completed, as shown in Table 3.

## 4. Flight Phase Motion Experiment

We compare the torso center of mass trajectory, torso center of mass velocity, ground reaction force in the transition section, and motion energy during the flying phase motion of the rigid-torso quadruped robot and the flexible-spine quadruped robot, and then we analyze the effect of the flexible spine on the flying phase motion. The BQR3 quadruped robot is taken as the research object, as shown in Figure 10. The length and mass of the torso of the BQR3 robot are treated according to the treatment of front–back = 3:2, corresponding to the simplified model shown in Figure 3.The parameters of each size and mass of the BQR3 robot itself are shown in Table 4. The specific initial values of the individual physical quantities of the flight phase motion process are shown in Table 5.

### 4.1. Trunk Center of Mass Trajectory

A comparison of the trajectories of the torso center of mass motion trajectories over time for a rigid-torso quadruped robot and a flexible-spine quadruped robot with different spine initial joint angles at different initial velocities is shown in Figure 11. The dashed line in the figure indicates the take-off motion, and the solid line indicates the flying phase motion.

From a single graph, it can be seen that the initial joint angle of the flexible spine affects the position of the total center of mass of the torso; the larger the absolute value of the initial angle, the lower the position of the center of mass at the take-off point. On the other hand, the larger the absolute value of the initial angle, the faster the rise of the center of mass position in the take-off motion, the higher the position of the take-off point, and the higher the highest position of the center of mass during the whole motion.

Combining the multiple graphs, it can be seen that the horizontal distance moved by the center of mass in the take-off motion does not change significantly with increasing speed for the same spine angle, but the horizontal distance moved by the center of mass in the flying phase motion increases significantly. The flexible spine can not only enhance the height of the center of mass movement, but also enhance the horizontal distance moved by the center of mass when the velocity is larger, which can help the robot achieve a higher and farther flying phase trajectory.

### 4.2. Trunk Center of Mass Velocity

The trajectories of the horizontal and vertical velocities of the torso center of mass of the rigid-torso quadruped robot compared with those of the flexible-spine quadruped robot with different initial joint angles of the spine at different initial velocities over time are shown in Figure 12. From the single plot, it can be seen that the appropriate initial spine joint angle can improve the horizontal velocity fluctuation due to touchdown in the transition section. From the results of multiple plots, it can be seen that compared with the rigid torso, in the case of low speed and low stiffness of the spine joints, the appropriate initial joint angle of the spine can improve the horizontal velocity of the center of mass at the moment of take-off. And in the case of high speed and high stiffness, the larger the initial joint angle of the spine, the greater the enhancement effect on the horizontal and vertical velocities at the moment of take-off.

### 4.3. Ground Reaction Force

A comparison of the ground reaction force on the rigid-torso quadruped robot and the flexible-spine quadruped robot with different initial joint angles of the spine at different initial velocities in the transition segment is shown in Figure 13. In order to quantify the magnitude of the ground reaction force more intuitively, the normalized ground reaction force is obtained by dividing the obtained ground reaction force by the gravity of the robot torso, and the graph is plotted with this quantity as the vertical coordinate. From a single graph, it can be seen that for the ground reaction force in the horizontal direction, the larger the initial joint angle of the flexible spine, the larger the peak force of the ground reaction force in both positive and negative directions. From the results of multiple plots, it can be seen that for the ground reaction force in the vertical direction, the ground reaction force in the vertical direction fluctuates based on the ground reaction force curve corresponding to the rigid torso as the spine joint angle increases at low speed. In the high-speed case, the impulse of the vertical ground reaction force increases with the increase in the initial spine joint angle. The same is true for the ground reaction force in the horizontal direction. Therefore, the spinal joints will have an effect on the ground reaction force. At low speeds and low stiffness, the effect on the horizontal ground reaction force is more pronounced; at high speeds and high stiffness, there is an effect on the ground reaction force in both directions, and a flexible spine will result in a greater value of the foot–ground contact force relative to a rigid spine, with a greater angle of the spinal joint having a greater effect.

## 5. Discussion and Conclusions

In this paper, we focused on the investigation of the impact of a flexible spine on the aerial phase motion of quadruped robots. We conducted dynamic modeling and motion characteristic analysis on a quadruped robot. Inspired by the aerial phase motion in biological systems, we used Fourier series to fit the ground reaction forces acting on the robot and described the parameterized aerial phase motion for both rigid- and flexible-spine quadruped robots. Employing a comparative analysis approach, we set different initial values for the parameterized aerial phase motion of the robot and compared the motion data between rigid- and flexible-spine quadruped robots. We observed differences in the robot’s performance in terms of the trajectory of the trunk’s center of mass, the velocity trajectory of the trunk’s center of mass, and the ground reaction forces during the transition phase. We analyzed the impact of a flexible spine on the aerial phase motion of quadruped robots. The results showed that a flexible spine could increase the upward speed during the transition phase, achieve a higher vertical position at the take-off point, and result in a higher maximum height during the aerial phase. In high-speed scenarios with high spinal stiffness, the flexible spine enabled the robot to cover a longer horizontal distance during the aerial phase. For initial joint angles of moderate size, the flexible spine improved the negative work on horizontal velocity caused by ground reaction forces at the beginning of the touchdown, reducing the decrease in horizontal velocity and minimizing its fluctuation. The flexible spine had an impact on ground reaction forces, generally leading to larger values in the corresponding positive or negative direction. In the future, we plan to use optimization methods to consider the position and attitude trajectories of the center of mass and plan the ground reaction forces during the transition phase comprehensively. Additionally, we intend to analyze the short aerial phase from the front leg touchdown to the rear leg touchdown in a similar manner and integrate the results of both analyses to make the analysis of the motion process more complete and the results more convincing. In the next steps of our work, we aim to validate the results using simulations or physical experiments.

## Figures and Tables

**Figure 1 biomimetics-09-00132-f001:**
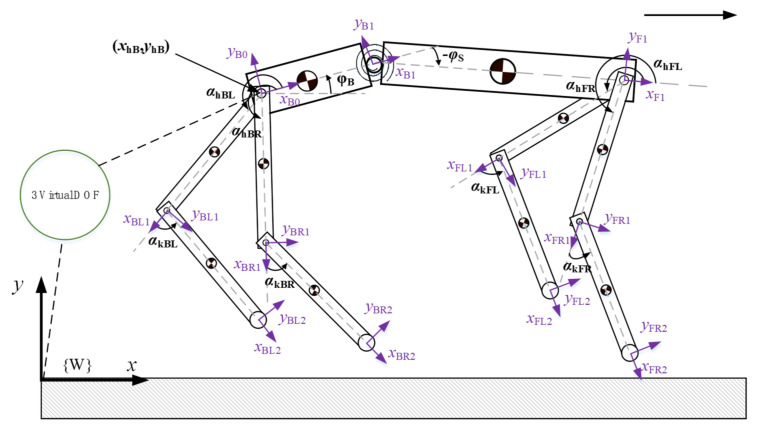
Quadruped robot model with a spine.

**Figure 2 biomimetics-09-00132-f002:**
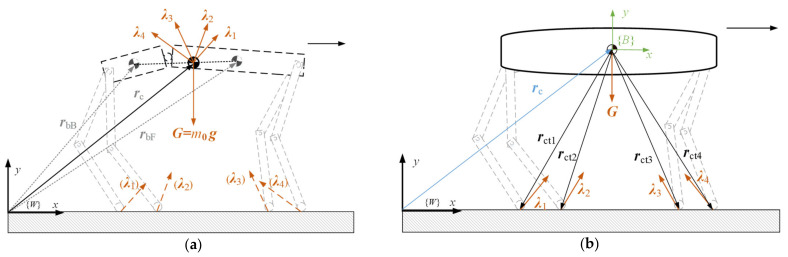
Model of quadruped robot. (**a**) Point mass model. (**b**) Single-rigid-body model.

**Figure 3 biomimetics-09-00132-f003:**
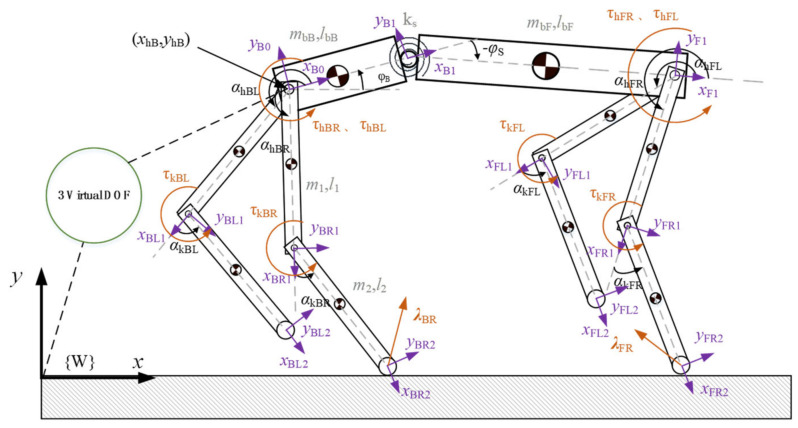
Planar multiple-rigid-body model.

**Figure 4 biomimetics-09-00132-f004:**
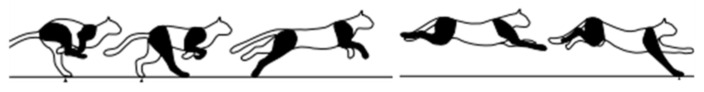
Take-off motion and flight phase motion.

**Figure 5 biomimetics-09-00132-f005:**
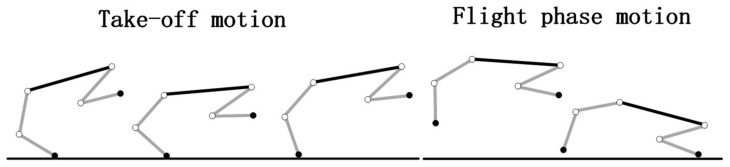
The process of quadruped robot with rigid torso.

**Figure 6 biomimetics-09-00132-f006:**
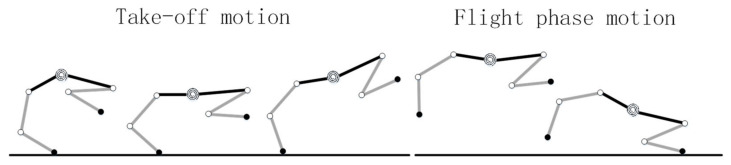
The process of quadruped robot with a flexible spine.

**Figure 7 biomimetics-09-00132-f007:**
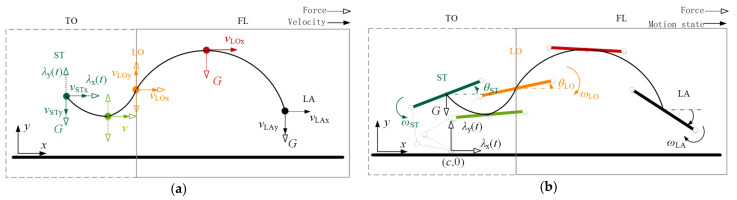
The flight phase motion of a rigid-torso quadruped robot. (**a**) Translational motion of a rigid-trunk quadruped robot. (**b**) Rotation of a rigid-torso quadruped robot.

**Figure 8 biomimetics-09-00132-f008:**
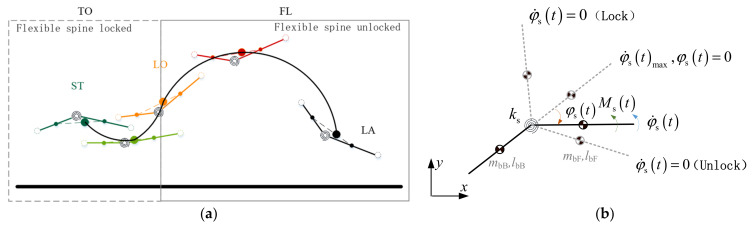
The flight phase motion of a flexible-spine quadruped robot. (**a**) The diagram of motion of the quadruped robot with a flexible spine. (**b**) Motion process of the flexible spinal joint.

**Figure 9 biomimetics-09-00132-f009:**
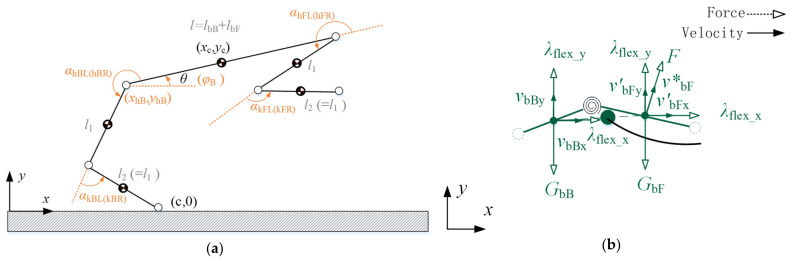
(**a**) Geometric relationship between flight parameters and generalized coordinates. (**b**) Velocity analysis of the total centroid of the flexible spine in take-off motion.

**Figure 10 biomimetics-09-00132-f010:**
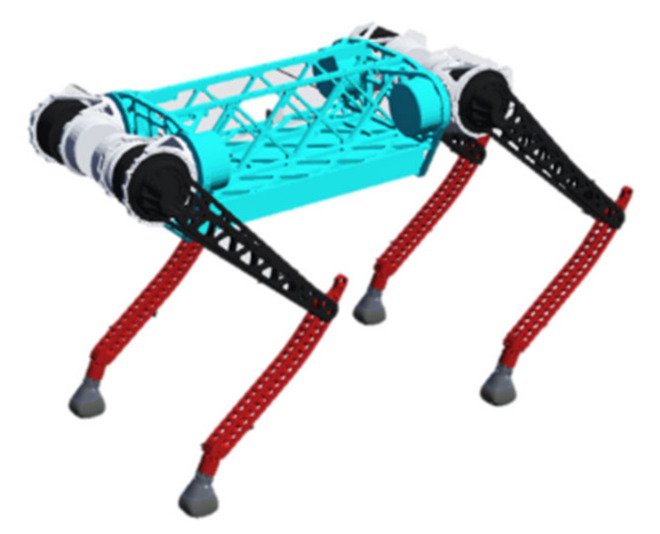
BQR3 quadruped robot.

**Figure 11 biomimetics-09-00132-f011:**
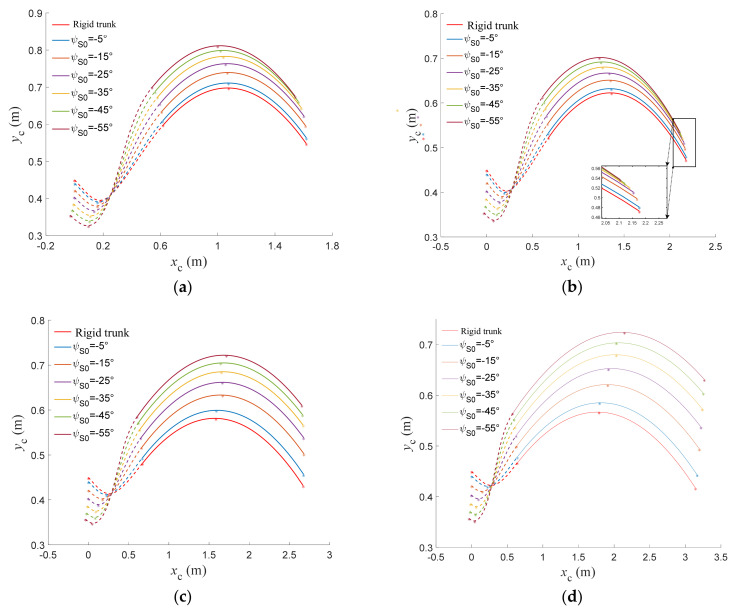
Trunk center of mass trajectory with different initial velocities. (**a**) The initial velocity is 2.5 m/s. (**b**) The initial velocity is 3.5 m/s. (**c**) The initial velocity is 5.2 m/s. (**d**) The initial velocity is 7.0 m/s.

**Figure 12 biomimetics-09-00132-f012:**
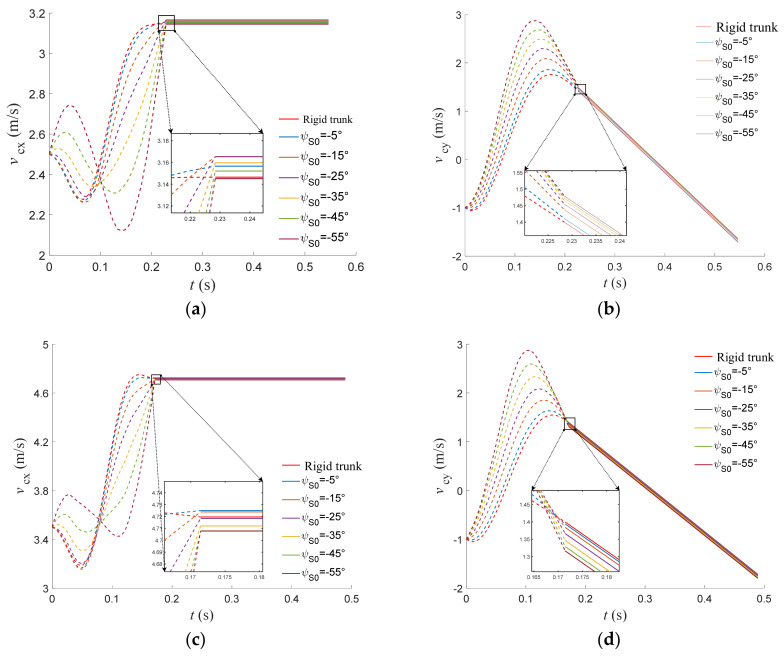
Horizontal and vertical velocity trajectories of trunk centroid with different initial velocities. (**a**) Horizontal velocity in x direction (2.5 m/s). (**b**) Vertical velocity in y direction (2.5 m/s). (**c**) Horizontal velocity in x direction (3.5 m/s). (**d**) Vertical velocity in y direction (3.5 m/s). (**e**) Horizontal velocity in x direction (5.2 m/s). (**f**) Vertical velocity in y direction (5.2 m/s). (**g**) Horizontal velocity in x direction (7.0 m/s). (**h**) Vertical velocity in y direction (7.0 m/s).

**Figure 13 biomimetics-09-00132-f013:**
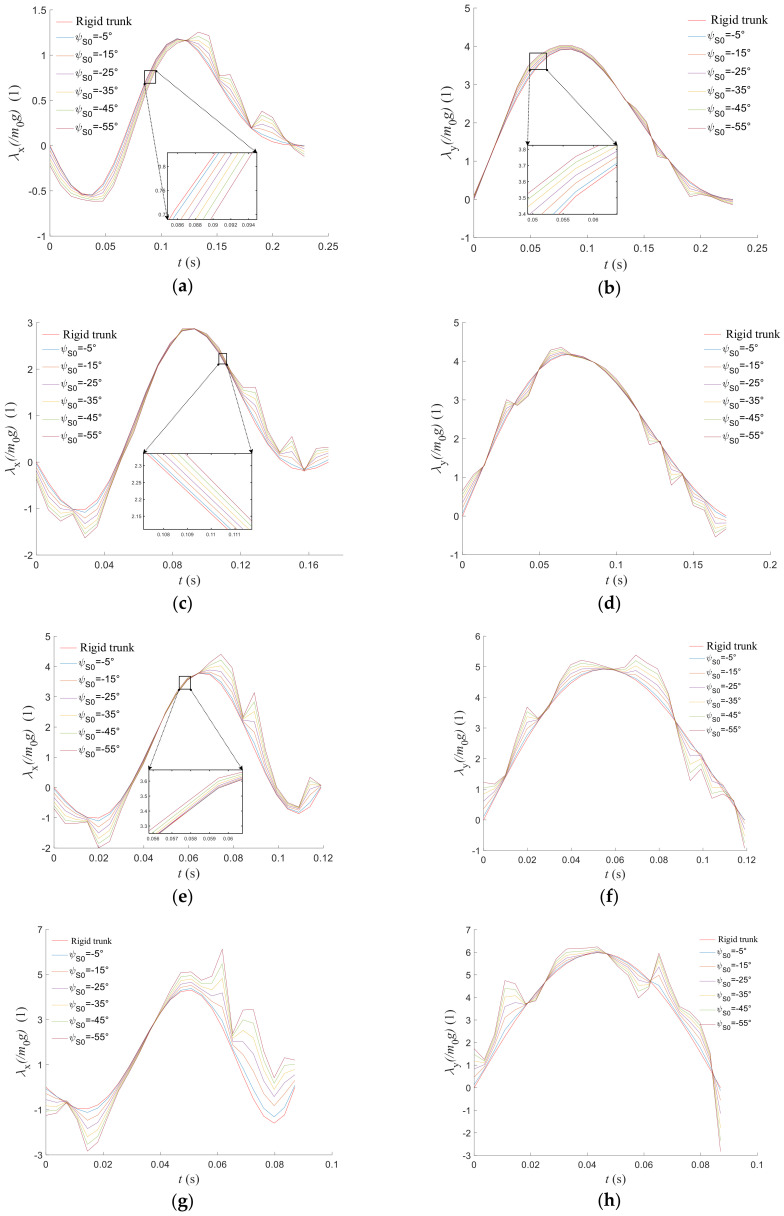
The ground reaction force of a quadruped robot with different initial velocities. (**a**) Ground reaction force in x direction (2.5 m/s). (**b**) Ground reaction force in y direction (2.5 m/s). (**c**) Ground reaction force in x direction (3.5 m/s). (**d**) Ground reaction force in y direction (3.5 m/s). (**e**) Ground reaction force in x direction (5.2 m/s). (**f**) Ground reaction force in y direction (5.2 m/s). (**g**) Ground reaction force in x direction (7.0 m/s). (**h**) Ground reaction force in y direction (7.0 m/s).

**Table 1 biomimetics-09-00132-t001:** The parameters of this model.

Parameter	Measure	Definition
$x_{hb},y_{hb}$	m	Back hip joint position
$\varphi_{B}$	rad	Hindquarters pitch
$\varphi_{s}$	rad	Trunk angles
α	rad	Joint angles

**Table 2 biomimetics-09-00132-t002:** The initial conditions for describing the trajectory of trunk.

Parameter	Measure	Definition
h	m	Altitude of flight phase
$\left(x_{LA},y_{LA}\right)$	m	Landing point position
$\left(v_{STx},v_{STy}\right)$	m	Starting point velocity
$\left(x_{ST},y_{ST}\right)$	m	Starting point position
(c,0)	m	Foot position
$\omega_{ST}$	rad/s	Starting point angular velocity
$\theta_{LA}$	rad	Landing point attitude angle

**Table 3 biomimetics-09-00132-t003:** The initial conditions for describing the position of the total centroid of the flexible spine.

Parameter	Measure	Definition
$\varphi_{s0}$	rad	Angle of the spinal joints
$\left(x_{ST\_flex},y_{ST\_flex}\right)$	m	Starting point position
$\left(v_{STx\_flex},v_{STy\_flex}\right)$	m/s	Starting point velocity

**Table 4 biomimetics-09-00132-t004:** The parameters of the BQR3 body.

Parameter	Value	Definition
$m_{bB}$	10.72 kg	Posterior trunk mass
$m_{bF}$	16.08 kg	Anterolateral trunk mass
$m_{1}$	2.6 kg	Femur link mass
$m_{2}$	0.81 kg	Tibia link mass
$l_{bB}$	0.24 m	Posterior trunk length
$l_{bF}$	0.36 m	Anterolateral trunk length
$l_{1}$	0.42 m	Femur link length
$l_{2}$	0.42 m	Tibia link length

**Table 5 biomimetics-09-00132-t005:** The parameters and values that need to be set.

Parameter	Value	Definition
h	0.1 m	Flight phase trajectory height
$\left(x_{LA},y_{LA}\right)$	(1, −0.05), (1.5, −0.05) (2, −0.05), (2.5, −0.05)	Landing point position
$\left(v_{STx},v_{STy}\right)$	(2.5, −1), (3.5, −1) (5.2, −1), (7, −1)	Starting point velocity
$\left(x_{ST},y_{ST}\right)$	(0, 0.45)	Starting point position
c	0.155, 0.132, 0.161,0.182	Touching point position
$\omega_{ST}$	0	Starting point angular velocity
$\theta_{LA}$	−15°	Landing point attitude angle
$\varphi_{s0}$	−5°, −15°, −25°,−35°, −45°, −55°	Angle of the spinal joints

## Data Availability

The data is contained within the article.

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
