# Peer review of "Research on Dynamic Modeling Method and Flying Gait Characteristics of Quadruped Robots with Flexible Spines"

_biomimetics, 2024, doi:10.3390/biomimetics9030132_

Round 1

Reviewer 1 Report

Comments and Suggestions for Authors

This paper investigated the impact of a flexible spine on the aerial phase motion of quadruped robots. The authors conducted many analyses to compare the performance of quadruped robots with rigid and flexible spines. Overall, the paper is well written and has solid contribution. However, several minor issues should still be addressed to further improve its quality. Below are some comments for the authors to consider:

1. In the experiments (section 4), the BQR3 quadruped robot was utilized as the research object. However, according to Figure 10, it seems that this robot only has rigid trunk. So how can the authors use this robot to analyze the performance of flexible spines? Please clarify.

2. The authors are recommended to add some photos (in the paper or as supplementary material) to intuitively show the performance difference between quadruped robots with rigid and flexible spines.

3. The literature study part could be further extended. Currently, there are also other studies developing flexible (soft) legs to improve the performance of quadruped robots (see the references below). From this perspective, the authors are also recommended to mention those work in the Introduction section as important state of the art. Below are some related work on flexible (soft) legs for quadruped robots:

“Design of topology optimized compliant legs for bio-inspired quadruped robots”. https://doi.org/10.1038/s41598-023-32106-5

"A soft quadruped robot enabled by continuum actuators". https://doi.org/10.1109/CASE49439.2021.9551496

Reviewer 2 Report

Comments and Suggestions for Authors

The authors developed dynamic models for quadruped robots equipped with a flexible spine, simulating the flight phase motion observed in cheetahs during high-speed galloping through parameterized process. And this manuscript compared the torso center of mass trajectory, velocity, ground reaction force in the transition section, and motion energy in flight phase of both rigid-bodied and flexible-spine quadruped robots under various initial conditions. The models, methods, results, discussions are stated in detail. The research provides critical insights for future quadruped robot design.

I appreciate the opportunity to review this manuscript. Overall, I find this manuscript is well-written and their research was well-conducted. I think this manuscript can be accepted in the current form.

Author Response

Thank you very much for reviewing our paper and for your recognition and valuable comments. We are very pleased that you found our research worthy of publication and thank you for recognizing our work.

We appreciate the feedback you provided during the review process, which is important for us to further improve the paper. We will carefully consider your suggestions and make the necessary changes in the final version to enhance the quality and readability of the paper.

Thank you again for your support and encouragement, and we look forward to further improving our research under your guidance.

Reviewer 3 Report

Comments and Suggestions for Authors

The paper presents the dedtailed investigation on the impact of the flexible spine on the motion of quadruped robots with comparison with rigid spine robots. Authors focused on the flight phase of the motion and provided mathematical models for its dynamics for robots with flexible and rigid spines and studied its behavior with different initial conditions. Current research can be a good base for further more specific motion control studies related to quadruped robots. The paper is goo for publishing.

Author Response

(The authors gave the same response as above.)

Round 2

Reviewer 1 Report

Comments and Suggestions for Authors

No further comments.